# Phytate Content in Cereals Impacted by Cropping System and Harvest Year

**DOI:** 10.3390/foods14030446

**Published:** 2025-01-29

**Authors:** Mailiis Korge, Maarika Alaru, Indrek Keres, Kaidi Möll, Liina Talgre, Ivo Voor, Illimar Altosaar, Evelin Loit-Harro

**Affiliations:** 1Institute of Agricultural and Environmental Sciences, Estonian University of Life Sciences, Kreutzwaldi 5, 51006 Tartu, Estonia; 2Proteins Easy Corp, 75 Campus Drive, Kemptville Agricultural College Campus, Kemptville, ON K0G1J0, Canada

**Keywords:** organic, conventional, temperature, cereal grains, phytic acid, spring barley, winter wheat

## Abstract

Phytate is a substance that has been considered mainly as an antinutrient, but at the same time it is a significant source of phosphorus and has several useful health-related properties that could be exploited. In this respect, a field experiment was conducted to study the effect of organic and conventional cropping systems with nitrogen (N) and phosphorus (P) amounts from 0 to 150 kg ha^−1^ and 0–25 kg ha^−1^, respectively, in six years (2017–2022) of weather conditions on phytate content in Estonia. Winter wheat had a higher phytate content of 1.9 ± 0.13 g 100 g^−1^ compared to spring barley with 1.1 ± 0.05 g 100 g^−1^. Fertilization with N or P did not affect phytate content in grains. Harvest year weather conditions (precipitation and air temperature) had a strong effect on phytate content. at a specific stage of plant development. Higher values of growing degree days in June and July, which sum in the experimental period varied between 609 and 978 °C, increased phytate content in winter wheat grains (flowering and grain filling stage), while the impact on spring barley phytate content was opposite (heading and flowering stage). Future research should study phytate content in grains grown on varying fertility level soils.

## 1. Introduction

Phytate (*myo*-inositol hexaphosphate, InsP6), also known as the salt of phytic acid, is a widespread substance in the plant kingdom serving as storage of phosphorus (P) and minerals. It is also the most common form of organic P in soil [1]. Phytate is found in legumes and nuts but is most abundant in cereals, where its content ranges from 0.06 to 2.2% [2]. In most cereal seeds, phytate is primarily located in the aleurone layer, where it supplies P to the growing plant embryo during seed germination [3]. Plants store 65–85% of their total P reserves as phytate, which constitutes 1–2% of the seed weight [4,5,6].

Phytate is traditionally considered a non-nutritional compound [7] or even an antinutrient due to its ability to form complexes with proteins and certain essential micronutrients, such as iron and zinc, significantly reducing their bioavailability [5,8,9]. Consequently, consuming phytate-rich foods can lead to micronutrient deficiencies in the body, although in low quantities it can display beneficial properties, like antioxidant [10], antidiabetic, and antibacterial effects [2,5].

The biosynthesis of phytate begins soon after anthesis, and its accumulation in the grains continues until seed maturation and drying [11]. This is similar to grain dietary fibers, arabinoxylans (AX) and beta-glucans (BG), which are known for their beneficial health-promoting properties. Previous studies of this current long-term field trial have determined the content of dietary fibers, AX and BG, in winter wheat and spring barley [12,13]. Still, the associations between phytate with AX and BG, which may have similar accumulation patterns, are unexplored. Previous studies have indicated that phytate biosynthesis is influenced by various factors, including fertilization and environmental conditions, such as weather [6,10]. Experiments have shown that P fertilization has a positive effect on the P content of oats [14] and rice grain [4]. Fukushima et al. [4] stated that rice phytate content can be increased by P fertilization, but it depends on genotype. Such information on wheat and barley is lacking, especially when comparing winter and spring cereal crops. Saastamoinen [14] found that while both nitrogen (N) and P affect the phytate content of oat grains, N has a more significant impact. A study on rice revealed that increasing N fertilization levels up to 300 kg ha^−1^ reduced the phytate content in grains by 17% [15]. In a study on maize, Kaplan et al. [6] found that N fertilizer levels of 200 and 300 kg ha^−1^ significantly increased the phytate content of maize grains compared to the 100 kg ha^−1^ level, while varying water availability during the growing period did not affect grain phytate content. Phytate content is also influenced by the cultivated crop species, ranging from 0.01% in mango to 2.2% in wheat [10] and by the selected genetic variety for cultivation [15].

Both winter and spring cereals constitute a significant part of the crop production sector in temperate climates, making it crucial to understand the phytate content variation in cereals. Just as long as the use of P from rock P continues at the current rate, agriculture is heading towards a P fertilizer shortage. As European agricultural goals include the reduced reliance on synthetic fertilizers and the more efficient use of waste stream nutrients [16], it is important to understand the impact of organic and conventional fertilization on grain components, including phytate. In the case of organic farming, when solid organic fertilizer is used, the absorption of phytate-P by plants is difficult and very sensitive to weather conditions [17]. The present field experiment compares the effects of mineral and organic N and P on winter and spring cereal phytate content. The aim of this study was (i) to analyze the combined effect of organic or conventional fertilization and annual weather conditions on the phytate content of winter wheat and spring barley and (ii) relationships between phytate content and quality values of these cereal grains.

## 2. Materials and Methods

### 2.1. Experimental Setup

This study was based on a two-factor long-term field experiment at the Estonian University of Life Sciences. The first trial factor was N and P fertilization treatment in conventional and organic systems; the other factor was trial year, i.e., air temperature and precipitation. The research focused on winter wheat (*Triticum aestivum* L.) ‘Fredis’ and spring barley (*Hordeum vulgare* L.) ‘Anni’, grown under both organic and conventional cropping conditions with or without different organic and conventional N and P fertilization levels over a six-year period (2017–2022) on *Stagnic luvisol* soil.

The conventional system treatments (Table 1) differed in the amounts of mineral nitrogen (NH_4_NO_3_) applied: N0 (control, N_0_P_0_K_0_), N1 (N_40-50_P_25_K_95_), N2 (N_80-100_P_25_K_95_), and N3 (N_120-150_P_25_K_95_), while P and K amounts (YaraMila CropCare 3-11-24 (Yara, Oslo, Norway)) per hectare were at the same level. Barley with undersown red clover was fertilized with lower N rates (N_40_, N_80_, N_120_), while winter wheat received higher N rates (N_50_, N_100_, N_150_).

The organic system treatments (Table 1) included control (Org0), winter cover crops (Org1), with nutrients incorporated into the soil in spring, and winter cover crops combined with composted cattle manure. Manure was applied manually to winter wheat in spring as a topdressing on the surface layer of the soil after the snow melted and plowed into the soil for spring barley before sowing (10 t ha^−1^; Org2). N, P, and K amounts applied with manure depending on the dry matter in different trial years were 36–66, 9–12, and 40–66 kg ha^−1^, respectively. Winter cover crops, a mixture of winter rye (*Secale cereale* L.) and winter oilseed rape (*Brassica napus ssp. oleifera var. biennis*), were used after the harvest of winter wheat and winter rye before the sowing of barley for the purpose of green manure. Cover crops were plowed into the soil after the snow melted in April. Based on P amounts applied, the experiment compared three P fertilization levels: P0 in N0, Org0, and Org1 with P 9–12 kg ha^−1^ in Org2 and P 25 kg ha^−1^ in N1, N2, and N3 treatments.

### 2.2. Sampling, Analysis, and Weather

Each year in mid-April, before the start of the fieldwork, soil samples were taken from each test plot to a depth of 20 cm. Soil pH was determined from 2 mm sieved, air-dried samples in 1 M KCl 1:2.5 solution. Acid digestion with sulfuric acid solution was used to determine the P_tot_ and K_tot_ concentrations of cattle manure. Total nitrogen (N_tot_) content of the oven-dried composted manure samples was determined by the dry combustion method with a varioMAX CNS elemental analyzer (ELEMENTAR, Langenselbold, Germany). The available P and K concentrations in soil samples were determined using the ammonium lactate (AL) method [18]. Soil nutrient contents averaged over experimental years and treatments were as per the following: N 1.23 g kg^−1^; P 0.11 g kg^−1^; K 0.15 g kg^−1^; C 142 g kg^−1^; and pHKCl, 5.84.

To determine the phytic acid content in grain samples, the P released by phytase and alkaline phosphatase was measured following the protocol of the Phytic Acid (phytate)/Total Phosphorus Assay Procedure (Megazyme International Ltd., Wicklow, Ireland). The measurements were conducted within the limits of the protocol used on the assumption that acid extraction of inositol phosphates and phytase treatment should ensure the release of phosphate from most of the different myo-inositol phosphate isoforms. The optical density of the measured solution was read at 655 nm using a UV-Vis spectrophotometer, Nanodrop One/One (Thermo Fisher Scientific, Waltham, MA, USA). In milled samples (Perten Laboratory Mill 3100 (Perten Instruments AB, Hägersten, Sweden)), the content of arabinoxylan (AX) and beta-glucan (BG) was determined as described in Korge et al. [12] and Khaleghdoust et al. [13], respectively. Grain yield (t ha^−1^) and quality parameters, 1000 kernel weight (1000 KW), test weight (g l^−L^), and protein content (%) were calculated as an average of four replications.

Information on the weather conditions for the field experiment was collected from the iMETOS 3.3 weather station (Pessl Instruments GmbH, Weiz, Austria) located near the Eerika experimental field (Table 2). The text also uses the expression ’year’, which expresses the combined effect of temperature and precipitation. In the calculation of growing degree days (GDD), the base temperature was +5 °C. The BBCH scale [19] for cereals was used to determine the development stages of winter wheat and spring barley; determination was made by observation (Table 3).

### 2.3. Statistical Analysis

All measurements of phytic acid content were performed in triplicate. The results are presented as means ± standard error. Statistical analyses were conducted using the Statistica 13 software package (TIBCO Software Inc., Palo Alto, CA, USA) through one-way and multifactor analysis of variance (ANOVA). Fisher’s Least Significant Difference (LSD) post hoc test was used to determine the differences between means when significant differences (*p* < 0.05) were found. The strength of correlations was expressed using Pearson’s correlation coefficient (r).

## 3. Results

In general, the phytate content of winter wheat was significantly higher than that of spring barley, whereas the mean over trial years and fertilizing treatments was 1.9 ± 0.13 for winter wheat and for barley only 1.1 ± 0.05 g 100 g^−1^. In this two-factorial trial, the proportion of variance for weather conditions and fertilizer treatment for winter wheat was 95% and 3%, and for spring barley, 53% and 1%, respectively. The content of phytate in winter wheat and spring barley was influenced significantly only by weather conditions, i.e., precipitation and temperature.

### 3.1. Impact of Cropping System on Phytate Content

The mean phytate content of winter wheat over trial years in the organic system (average of Org0 + Org1 + Org2) and the conventional system (average of N0 + N1 + N2 + N3) was 2.1 ± 0.19 and 1.9 ± 0.14 g 100 g^−1^, respectively, which did not differ significantly from each other. The coefficient of variation in both systems was statistically the same, i.e., 38% and 36%, respectively. The mean phytate content over trial years in spring barley organic and conventional systems was 1.1 ± 0.07 and 1.2 ± 0.05 g 100 g^−1^, respectively, which also did not differ significantly from each other. The coefficient of variation in the barley organic system was significantly higher than that of the conventional system, i.e., 27% and 21%, respectively.

Comparing the phytate content of grains in the unfertilized treatments with those that received an additional 9−12 and 25 kg ha^−1^ of P (Org2, N1, N2, N3), no significant changes in phytate content were observed for either cereal. The phytate content in barley was 1.15 ± 0.07 g per 100 g^−1^ in the unfertilized treatments compared to an average of 1.17 ± 0.10 g per 100 g^−1^ in Org2 and 1.14 ± 0.06 g per 100 g^−1^ in the N1, N2, and N3 treatments. In wheat, the corresponding values were 2.07 ± 1.15, 2.02 ± 1.17, and 1.85 ± 1.14 g per 100 g^−1^, respectively (Figure 1).

### 3.2. Effect of Weather Conditions on Phytate Content

While the variability in wheat phytate content among fertilization treatments was 0.4 g 100 g^−1^, the variability between years was as high as 2.1 g 100 g^−1^. The phytate content of winter wheat and spring barley differed significantly during trial years (Figure 2). The phytate content of winter wheat ranged from 1.32 (in 2020) to 3.39 (in 2021) g 100 g^−1^; the mean over years and treatments was 1.97 g 100 g^−1^, while in spring barley, it ranged from 0.81 (in 2021) to 1.42 (in 2017) g 100 g^−1^. The mean over years and treatments was 1.15 g per 100 g^−1^. While years with the lowest phytate content in wheat (2017, 2020) had values below 1.5 g 100 g^−1^, then years 2021 and 2022 stood out with significantly higher phytate content, exceeding content of 2 and even 3 g 100 g^−1^.

During the vegetation period, the phytate content of the studied cereals was most affected by the temperature and precipitation in June and July, while this effect on the phytate content of wheat and barley was opposite (Figure 3a,b).

Phytate content of winter wheat grains in this period correlated positively with temperature (*p* < 0.001) and negatively with precipitation (*p* < 0.001); however, spring barley phytate correlated negatively with temperature in June (*p* < 0.001) and July (*p* < 0.05) and positively with precipitation in June (*p* < 0.01) (Table 4). The contrary effect of weather on the phytate content of winter wheat and spring barley during the experimental years was surprising. The greatest impact was observed for GDD and the amount of precipitation in June and July, i.e., during the period when winter wheat went through the development stages from booting (BBCH 47) to harvest maturity (BBCH 89), while barley lagged behind in its development about two weeks (Table 3). The sum of June and July growing degree days (GDD) varied between 609 (in 2017) and 978 (in 2021) °C.

### 3.3. Relationship of Phytate Content with Quality Values of Cereals

Just as the variable weather during the growing season affected the phytate content of winter wheat and spring barley differently, the relationships between the phytate content and cereal quality indicators of barley and wheat grains were also variable (Figure 4). The mean 1000 kernel weight (1000 KW) of winter wheat over treatments varied between 34.8–49.3 g in the trial years, whereas, in contrast to the phytate content, the 1000 KW values were lower in recent years of the experiment. Barley 1000 KW did not correlate with phytate content to a significant extent, while phytate content in barley grains was greater with higher yield, which varied from 1.3 to 3.8 t ha^−1^ over treatments (*p* < 0.05). The grain yield of winter wheat was 2.9–4.5 t ha^−1^ over treatments; grain yield and phytate content of winter wheat had no significant correlation. Mean test weight values of winter wheat over treatments varied between 738–816 g L^−1^, and phytate contents in grains were higher in cases where test weight values of wheat were close to 780 g L^−1^ (i.e., in 2020–2021). Barley test weight values were typical for this grain, but there were no significant relationships with phytate content.

Arabinoxylan (AX) content of wheat and barley correlated positively with their phytate content (r = 0.67, *p* < 0.001 and r = 0.36, *p* < 0.05, respectively), and barley phytate content correlated negatively with beta-glucan (BG) content (r = −0.48; *p* < 0.001). The correlation with AX was stronger for wheat, where AX is the main dietary fiber located mostly in the aleurone layer, as the BG content was very low. Protein content had a significant correlation only with barley phytate content (r = −0.39, *p* < 0.05).

## 4. Discussion

Mineral P migrates in the biosphere from the rhizosphere into the field-to-fork paradigm. This long-term study of inositol-hexaphosphate aimed to monitor this partitioning dynamic by measuring phytate content in winter and spring cereal grains produced in different agricultural and climatic conditions. Based on six years of experimental data, the phytate content in winter wheat grains was significantly higher (*p* < 0.05) compared to spring barley, and the phytate content in wheat (1.97 ± 0.13 g per 100^−1^ g) was also significantly higher than the average of the range reported by Schlemmer et al. [2] (0.78 g per 100^−1^ g). That report summarized various authors findings in the period from 1976 to 2005, stating that the phytate content in wheat ranged from 0.39 to 1.35 g 100 g^−1^ and in barley from 0.38 to 1.16 g per 100 g^−1^. In this current experiment, winter wheat average phytate content was significantly higher in the last two experimental years (the average of 2021 and 2022 was 2.81 g 100 g^−1^), while the average from 2017–2020 was 1.55 g 100 g^−1^. This result refers to the possible effect of elevated air temperatures during grain filling, i.e., the period of phytate accumulation, which had a positive effect on grain phytate content. A rapid rise in June–July temperatures has been observed on the same experimental field over a 15-year period [20].

Several authors have found that N fertilizer increases the content of phytate in grains [4,6,14], while in this field trial the effect of N fertilizer on phytate content was insignificant, probably due to rather small quantities of N applied per ha. Organic and mineral P fertilization did not affect the phytate content in cereals in this current experiment. Nutrient uptake by crops takes place mostly during their vegetative growth with their accumulation in plant leaves, from where they are translocated to the grain [21]. Soil P uptake and translocation in the tillering and stem elongation stage are stimulated by the application of P fertilizer [4]. However, a previous analysis of P available to plants in the soil of the same experiment showed that the content of plant-available P in unfertilized treatments did not differ significantly from fertilized treatments [22]. When plants are growing in soil with optimal nutrient content, additional nutrients may not affect plant nutrient content, which is likely what occurred in the present field experiment.

The effect of the cropping systems as an average of treatments on phytate content was similar for winter wheat but significantly different for spring barley. The mean phytate content over barley treatments varied between trial years, with the organic system having a higher mean than the conventional system, 27% and 21%, respectively, whereas there was no plausible difference between the mean values of the two systems for winter wheat (38% and 36%, respectively). However, despite the lack of difference in phytate content between cropping systems for wheat, it should be noted that the coefficient of variation in wheat phytate content was 19% higher in the conventional and 9% higher in the organic system than that of barley. If we compare different methods of manure fertilization, i.e., in the case of wheat, manure was applied as top dressing during the tillering stage, and in the case of barley, manure was plowed into the soil before sowing, the variation in phytate content as an average of trial years in topfertilized winter wheat grains was more than twice as high as that of barley (44% and 20%, respectively). Chtouki et al. [23] found that the release of P from manure and its availability to plants largely depends on soil moisture. In this experiment, at the time of manure application after snowmelt during the last 10 days of April, the sum of precipitation ranged between trial years significantly from 3 mm up to 52 mm (Table 2), which may have caused variable P availability by winter wheat roots in the early stages [24]. In the case of barley, the manure was plowed into the soil ca to a depth of 10–20 cm, where the availability of nutrients is more stable because it is not so dependent on the weather. In the conventional system, mineral P was introduced into the soil with sowing, where P availability by plant roots was faster, which probably ensured a more stable phytate content of barley in this system.

From the two studied factors, the content of phytate in winter wheat and spring barley grains was significantly influenced only by weather conditions (*p* < 0.001). Weather conditions during the tillering and grain filling stages of the plants were the most relevant [20]. The variability of the phytate content of winter wheat in different experimental years as an average of cropping systems and treatments was considerably higher than that of barley; the coefficient of variation was 39% and 18%, respectively. The high coefficient of variation for winter wheat was caused by the high phytate values of 2021. The interaction of the organic fertilizer application technology with the weather could also have caused greater variability, as winter wheat received manure as topdressing in the spring, with nutrients more susceptible to losses when compared to soil application.

However, the effect of precipitation and temperature on these cereals was completely opposite. The reason could be the fact that winter and spring cereals were compared, the development phases of which did not coincide in time, as winter wheat was two weeks ahead of barley in its development (Table 3). It was found that four of the six test years have sums of GDD values 150–375 °C higher than the long-term average (Table 2). It is characteristic that the spring months have become cooler in this particular Nordic region, but the summer months of June–August are significantly warmer (Table 2). Such a change in the weather has also significantly affected the results of this present experiment, where the weather during the years of the experiment has been the determining factor. Winter wheat harvesting time varied from July 22 to August 10 in different experimental years. In general, winter wheat was harvested at the end of July; however, in one experimental year (2017) with the lowest GDD, it was harvested in August. Spring barley harvesting time varied from August 5 (in 2021) to August 28 (in 2017), usually harvested during the first 10 days of August. Winter wheat phytate content had a very strong positive correlation with June + July GDD (*p* < 0.001), while spring barley correlation with GDD was negative (*p* < 0.001; Figure 2). Correlation analyses showed that in the last experimental years, the phytate content level in winter wheat grains increased, and in spring barley grains, it decreased significantly (r = 0.59, *p* < 0.001 and r = −0.48, *p* < 0.01, respectively). These results could have been caused by the higher GDD values in June and July of the last experimental years.

As for temperature, precipitation during the vegetation period had an opposite effect on the phytate content of wheat and barley (r = 0.47, *p* < 0.01 and r = −0.31, *p* < 0.05, respectively). At the same time there appeared an interaction between air temperatures and precipitation in connection to phytate accumulation. In addition to higher temperatures, the years 2021 and 2022 experienced the highest precipitation during the growing season. Fernando et al. [25] also found that higher water availability increases the phytate content in wheat grains. The years 2018 and 2019 with lower amounts of precipitation and less GDD may have inhibited phytate accumulation. Results show that the phytate content in wheat grains increased during growing periods with higher GDD in the case of sufficient amounts of precipitation. Conversely, this correlation did not hold for barley, where the highest phytate content was observed in 2017, a cooler growing season, and the lowest in 2021, the warmest and driest grain-filling period. It would be intriguing to further explore such an opposite effect of the weather on the phytate content of winter wheat and spring barley. One might assume that it was due to their different stages of development at a certain point in time.

The quality of the studied grain was also significantly affected by weather conditions. Among the grain quality indicators evaluated in the experiment, including grain dietary fibers for widening the quality parameter spectrum, in the case of barley, only its BG content had a significant relationship with the phytate content (r = −0.48, *p* < 0.01). The negative correlation was probably caused by the fact that phytate and β-glucan are deposited in different parts of the seeds (in the aleurone layer and endosperm, respectively). The phytate content of winter wheat was negatively correlated with 1000 KW and positively with AX content (Figure 3a,e). Aurangzeb et al. [26] came to a conclusion that low phytic acid genotypes of wheat exhibit improved agronomic traits, like seed size, after analyzing 12 wheat genotypes. Our results support this finding. Raboy [8] stated that low phytate content can be interpreted as an indicator of low yield and susceptibility to stress. In the case of barley, this was similar, as phytate content in grains was greater with higher yield. But in addition, results showed that 1000 KW of winter wheat was much lower in 2021 and 2022 when the temperature during the grain-filling period was much higher than the long-term average. This shortened the grain-filling period, and the movement of nutrients into the grains was inhibited, resulting in a higher percentage of phytate content in the seed. Phytate and AX in wheat are both located in the aleurone layer of grains, so there is a positive correlation between them [5].

## 5. Conclusions

Estonian cereals contained a considerable amount of phytate, typically ranging from 1 to 2 g 100 g^−1^. Based on the results of a six-year experiment, winter wheat had a higher phytate content compared to that of spring barley. The phytate content in grains was influenced solely by weather conditions, not by nitrogen or phosphorus fertilization or cropping methods. Lack of effect from P fertilization was probably due to the sufficient content of P in the soil, while the absent effect from N fertilization could have been a result of low N amounts applied. Grain phytate content could be affected by organic fertilizer application method, with manure topdressing being more vulnerable to weather conditions compared to manure being plowed into the soil. The two factors, precipitation and air temperature at certain plant development stages, did affect the phytate content in cereals. Higher air temperatures in June and July increased phytate content in winter wheat grains, while the impact on spring barley phytate content was opposite. From a practical point of view, it means that farmers aiming to produce lower phytate content grain can choose barley over wheat, but they cannot control the content with fertilization. For food producers and dietitians, it is important to acknowledge that phytate content depends on the annual weather conditions. Future research should up the ante and study phytate content in grains grown on soils with varying fertility levels. Furthermore, greater focus should go on exploring the valorization possibilities of phytate in cereals. 

## Figures and Tables

**Figure 1 foods-14-00446-f001:**
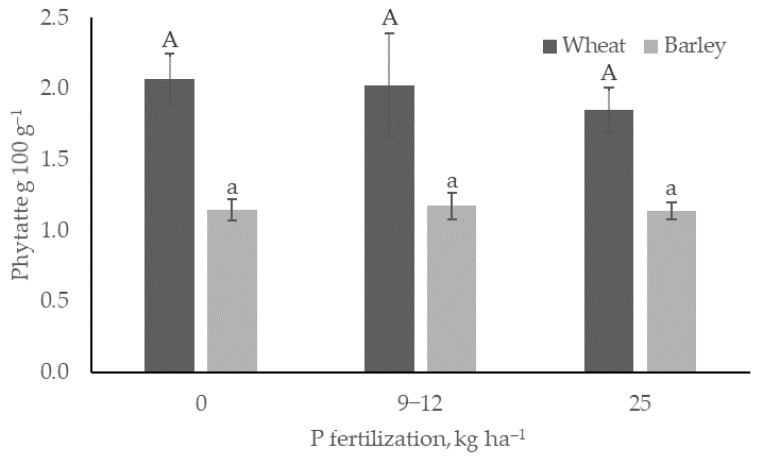
Average wheat and barley phytate contents depending on P fertilization level. Upper and lowercase letters on the bars indicate significant differences between treatments of the same crop species; bars with the same letter are not significantly different (*p* > 0.05).

**Figure 2 foods-14-00446-f002:**
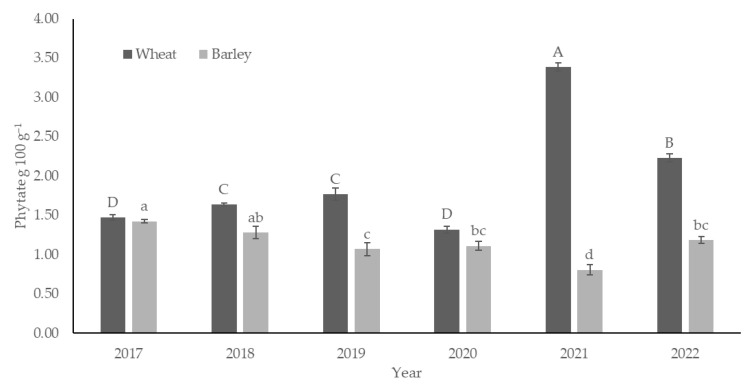
Phytate content (g 100 g^−1^) of winter wheat and spring barley in different experimental years as an average of fertilization treatments; n: letters on bars refer to comparisons between different cropping years of the same species; bars with the same letter are not significantly different (*p* > 0.05).

**Figure 3 foods-14-00446-f003:**
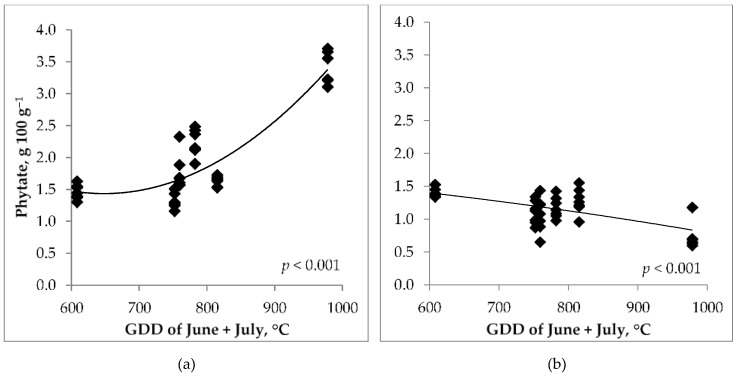
Correlations between wheat (**a**) and barley (**b**) phytate content and growing degree days (GDD) temperatures.

**Figure 4 foods-14-00446-f004:**
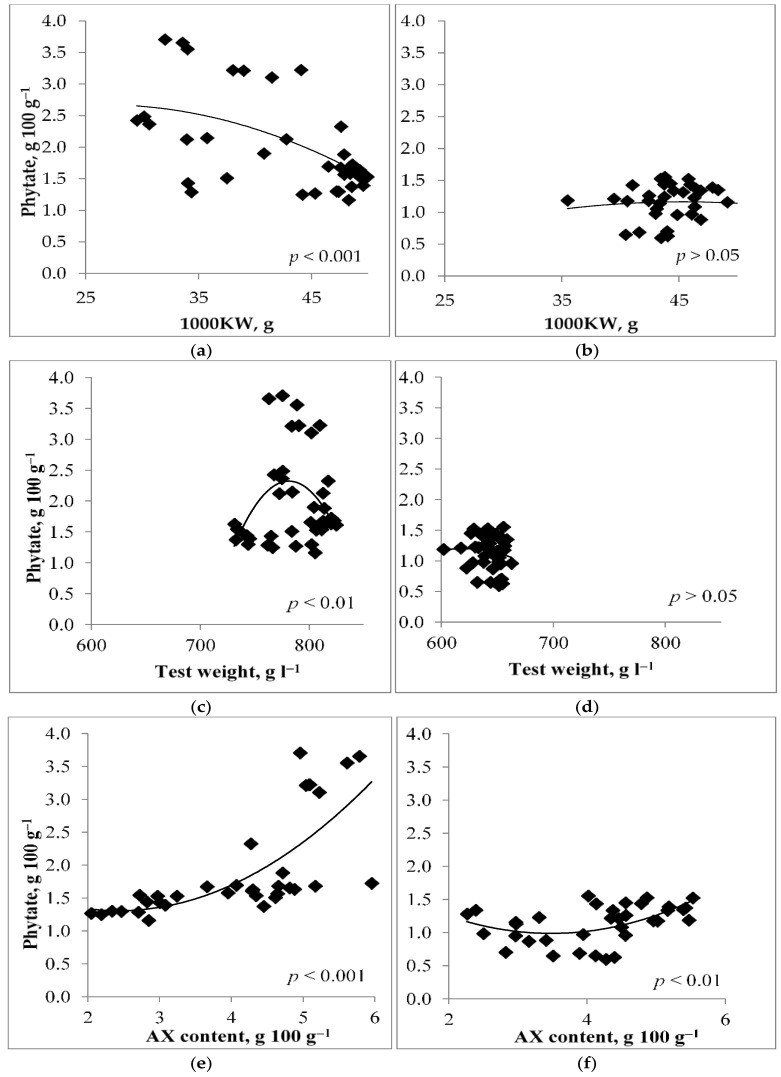
Correlations between wheat (**a**,**c**,**e**) and barley (**b**,**d**,**f**) phytate content and grain quality parameters.

**Table 1 foods-14-00446-t001:** Scheme of the field experiment organic (Org0–Org2) and conventional (N1–N3) treatments with replications (I–IV).

Organic	Org2 I Org1 I Org0 I	Org2 II Org1 II Org0 II	Org2 III Org1 III Org0 III	Org2 IV Org1 IV Org0 IV
intermediate strip				
Conventional	N3 I N2 I N1 I N0 I	N3 II N2 II N1 II N0 II	N3 III N2 III N1 III N0 III	N3 IV N2 IV N1 IV N0 IV

**Table 2 foods-14-00446-t002:** Average air temperatures (°C), sum of growing degree days (GDD) (°C), and monthly precipitation (mm) during the experimental period of April–August 2017–2022 and their long-term averages.

Month	Average Air Temperature per Month, °C						
	1991–2020	2017	2018	2019	2020	2021	2022
Average of April–August, °C	13.52	12.1	15.7	14.1	13.2	14.8	14.2
April	5.9	3.4	7.2	7.7	4.8	5.3	4.5
May	11.5	10.4	16.0	11.4	9.5	10.9	10.3
June	15.5	14.0	15.9	18.6	18.4	19.8	17.5
July	18.0	16.0	20.8	16.4	16.3	22.2	18.1
August	16.7	16.7	18.8	16.7	16.8	15.8	20.1
	Sum of GDD per month, °C						
Sum of April–August, °C	1295	1164	1669	1433	1284	1539	1445
April	50	16	85	109	24	40	28
May	201	177	341	204	141	187	166
June	314	269	327	407	401	445	375
July	386	340	488	352	351	533	406
August	344	364	429	361	366	334	469
	Sum of precipitation per month, mm						
Sum of April–August, mm	323	328	170	213	332	390	340
April	35	52	28	3	50	29	31
May	54	16	8	60	32	108	46
June	88	94	61	51	117	19	58
July	67	61	14	41	69	17	145
August	79	106	59	58	64	217	60

**Table 3 foods-14-00446-t003:** Development stages of winter wheat and spring barley (BBCH scale) by months in the April–August period.

	April	May	June	July	August
Winter wheat	21–22	23–45	47–71	72–90	x
Spring barley	x	0–30	31–65	66–85	85–90

**Table 4 foods-14-00446-t004:** Phytate correlations with cereal grain parameters and environmental factors; correlations given in bold are significant (*p* < 0.05).

Factor	Winter Wheat	Spring Barley
Grain quality parameters		
Yield t ha^−1^	−0.24	**0.41**
1000 KW	**−0.62**	−0.01
Test weight, g L^−1^	0.21	**−0.63**
Protein, %	0.01	**−0.39**
AX, g 100 g^−1^	**0.67**	**0.36**
BG g 100 g^−1^	0.09	**−0.48**
Air temperature, °C		
Average of the period	**0.42**	−0.31
April	0.05	−0.19
May	−0.05	0.19
June	**0.58**	**−0.71**
July	**0.75**	**−0.38**
August	**−0.51**	**0.43**
Precipitation, mm		
Sum of the period	**0.47**	**−0.31**
April	**−0.26**	**0.30**
May	**0.87**	**−0.69**
June	**−0.83**	**0.46**
July	**−0.62**	**0.28**
August	**−0.89**	**−0.45**

## Data Availability

The original contributions presented in the study are included in the article, further inquiries can be directed to the corresponding author.

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
