# Peer review of "Phytate Content in Cereals Impacted by Cropping System and Harvest Year"

_foods, 2025, doi:10.3390/foods14030446_

Round 1
Reviewer 1 Report
Comments and Suggestions for Authors
The manuscript titled “Phytate content in cereals impacted by cropping system and harvest year” investigates the effects of nitrogen and phosphorous fertilization and weather on cereal phytate content in field conditions during 2017–2022 in Estonia. My biggest concern is that the phytate released by phytase may not accurately represent the true phytate content. Below are specific comments for improvement:
1. Line 15–17: The fertilization and weather conditions should be described in more detail.
2. Line 19–24: More detailed and specific information is needed, including relevant data to support the statements.
3. Line 26: Some of the keywords are repeated in the title. Please revise.
4. Line 30: “myo” should be italicized.
5. Line 45–50, line 228-233: Why is the focus on the associations between phytate, AX, and BG? There seems to be no direct correlation between these parameters. Could you elaborate on their significance?
6. Line 53, 115, 117...: Since you’ve defined the abbreviations, it’s unnecessary to continue using the full terms. Please correct this.
7. Line 58–62: Is the variation in phytate content only due to different crop species? Also, how do the crop yields compare?
8. Materials and Methods: This section is too lengthy. Please consider dividing it into subsections for better clarity.
9. Line 86–90: Can you explain why these specific nitrogen levels were used in the study?
10. Line 115–117: The phosphorus content released by phytase is not equivalent to the actual phytate content in crops. Also, the total phosphorus content cannot be accurately measured by the phosphorus released by alkaline phosphatase.
11. Results: The actual total N/P and available N/P contents in soil should be reported.
12. Figure 3: Why do the effects of GDD on phytate contents differ between wheat and barley? Why was GDD used instead of average temperature?
13. Line 223: What do the “mean test weight values” refer to, and how are they calculated?
14. Line 226: Provide additional information about the N application to better explain the observed variations.
15. Line 269–271: This information seems redundant with the introduction. Please revise or remove.
16. Line 271–278: The P content does not influence phytate content in your study, so it may not be necessary to discuss it in detail.
17. Line 301–302: What about total P content? Does it affect phytate content in crops? Also, does fertilization or temperature affect P uptake in this study?
18. Conclusions: The conclusion should focus more on practical guidance for agricultural production.
Author Response
Response to Reviewer 1 Comments |
||
1. Summary |
|
|
Thank you very much for taking the time to review this manuscript. Please find the detailed responses below and the corresponding revisions/corrections highlighted/in track changes in the re-submitted files. |
||
2. Point-by-point response to Comments and Suggestions for Authors |
||
Comment 1. Line 15–17: The fertilization and weather conditions should be described in more detail. Comment 2. Line 19–24: More detailed and specific information is needed, including relevant data to support the statements. Response 1 and 2: The text of Abstract was revised to include more detail while still adhering to the journal guidelines limiting the abstract volume into 200 words. (lines 15-31). |
||
Comment 3. Line 26: Some of the keywords are repeated in the title. Please revise. Response 3: Keyword “phytate” (line 33) has been changed into a new word (phytic acid) not included in the title. |
||
Comment 4. Line 30: “myo” should be italicized. Response 4: Thank you for pointing this out. The suggestions have been implemented. |
||
Comment 5. Line 45–50, line 228-233: Why is the focus on the associations between phytate, AX, and BG? There seems to be no direct correlation between these parameters. Could you elaborate on their significance? Response 5: Thank you for pointing this out. As our idea was to understand the patterns and dynamics of changes in cereal grains phytate content depending on cropping type and fertilization rate, in addition our ambition was to find associations between phytate and grains quality parameters. As yield, 1000 kernel weight, test weight and protein are typical grain quality parameters we wanted to include also grain fibers, AX and BG, which may have similar accumulation patterns, due to their similar location in the grain. But these have jet been unexplored. Text has been supplemented to emphasize this goal (374-376) |
||
Comment 6. Line 53, 115, 117...: Since you’ve defined the abbreviations, it’s unnecessary to continue using the full terms. Please correct this. Response 6: Changes to the manuscript has been applied across the manuscript. |
||
Comment 7. Line 58–62: Is the variation in phytate content only due to different crop species? Also, how do the crop yields compare? Response 7: No, in addition to crop species there are studies that have shown the impact of fertilization on phytate content. In the work by Ning et al., 2009 with japonica rice they state a large reducing effect of N and a smaller effect of cultivar on grain phytic acid. At the same time, it was found that the grain yield progressively increased with N level. While Kaplan et al., 2019 states increasing effect of N rates on maize phytate content with some genotypic differences also. They stated also higher grain amylose content with higher N rates applied.
Based on those results it could be assumed that the accumulation of phytate in different crops has variable dynamics. Here is a major difference between rice and maize when talking about phytate, as in maize it accumulates in the germ, while in rice and other cereal grains in aleurone layer. This could be also the reason why in rice with increasing yield phytate content decreases, but in maize amylose content increases similarly to phytate. In our study phytate content in spring barley was in positive correlation with its yield, while with winter wheat there was no significant correlation. But rather there was negative association between wheat phytate and 1000 KW. Another reference on the matter has been added to the manuscript and the discussion has been revised (line 381-386)
|
||
Comment: 8. Materials and Methods: This section is too lengthy. Please consider dividing it into subsections for better clarity. Response 8: We agree, the changes has been made into the section of Materials and Methods by adding 3 sub-sections: experimental setup, sampling, analysis and weather and statistical analysis |
||
Comment 9. Line 86–90: Can you explain why these specific nitrogen levels were used in the study? Response 9: Phytate determination was conducted with samples collected from a long-term five field crop rotation comparison experiment established already on 2008. The experiment was established with 4 different N rates to cover control (N0) and low (40-50kg/ha), medium (80-100 kg/ha) and high (150 kg/ha) nitrogen fertilizer rates. The main idea was to compare N rates used in practical agriculture. |
||
Comment 10. Line 115–117: The phosphorus content released by phytase is not equivalent to the actual phytate content in crops. Also, the total phosphorus content cannot be accurately measured by the phosphorus released by alkaline phosphatase. Response 10: We thank this reviewer for raising this important aspect. We have revised the relevant text to take into account such caveat emptor. Please see new Lines (133 – 136). We agree that the 'available' phosphorus content released by phytase is not equivalent to the actual phytate content in crops. Also, the total phosphorus content cannot be accurately measured by the phosphorus released by alkaline phosphatase. The methods used yield approximations which are valuable first-indicators in their own right. As the Irish supplier states, " Megazyme has developed a simple, quantitative method (K-PHYT) to measure total “available phosphorus” released from food and feed samples that is amenable to high numbers of samples and does not require tedious anion-exchange purification." K-PHYT is a universally accepted assay (* Li et al., 2025; Karpinska & Foyer 2025). When determining phytate content in grains it is possible that not all of the analyte is released, reacted and determined. However, the K-PHYT method developed by Megazyme uses acid extraction of inositol phosphates. As such, phytase treatment should ensure the release of phosphate from most of the different myo-inositol phosphate isoforms. That the values obtained may reflect grain phytate levels is now qualified in the revised text. These current results were similar to or even higher than those reported earlier (e.g., Schlemmer et al., 2009). Thus, the Megazyme treatment may be an accurate and useful approach in this regard.
|
||
Comment 11. Results: The actual total N/P and available N/P contents in soil should be reported. Response 11: The content of total N and available P as an average of experimental years and treatments were N 1.23 g kg‒1; P 0.11 g kg‒1. However, there is no data available for available N and total P in soil. |
||
Comment 12. Figure 3: Why do the effects of GDD on phytate contents differ between wheat and barley? Why was GDD used instead of average temperature? Response 12: The accumulation of average daily temperatures or GDD for winter wheat and spring barley are different due to the crops different growing period. This is the main reason that gives also different correlations for both crop phytate content and GDD. We chose GDD instead of average temperatures, because GDD is well associated with crop growth stages, which we wanted to connect to the dynamics of cereal grains phytate content. |
||
Comment 13. Line 223: What do the “mean test weight values” refer to, and how are they calculated? Response 13: Thank you for pointing this out. Test weight value refers to the volumetric weight of grain. Information about yield, 1000KW, test weight and protein were added into the Material and methods section (lines 141-142). |
||
Comment: 14. Line 226: Provide additional information about the N application to better explain the observed variations. Response 14: The Material and Methods section 2.1. has been supplemented with more detail to make the manure application to both crops better understandable (lines 103-104). |
||
Comment 15. Line 269–271: This information seems redundant with the introduction. Please revise or remove. Response 15: The text was modified accordingly to the suggestions (lines 295-297). |
||
Comment: 16. Line 271–278: The P content does not influence phytate content in your study, so it may not be necessary to discuss it in detail. Response 16: Our result that phytate content was not affected by P fertilization was rather unexpected as previous studies have shown correlation. For this reason, we found it deserves some explanation. Indeed, it should not be in detail and the text was modified based on the reviewers’ suggestions. |
||
Comment: 17. Line 301–302: What about total P content? Does it affect phytate content in crops? Also, does fertilization or temperature affect P uptake in this study? Response 17: Available P influences P uptake by plants and their phytate accumulation. The amount of total P may also have effect, as it may determine also the amount of plant available P in the soil. From the analysis done before on the same experiment it seems that plant available P content does not depend on fertilization level of the current experiment (Keres et al., 2020). Air temperatures effect on plants phytate content was one of our aims in the study. And yes, based on six years data, it shows clearly that the grain P content depends on the conditions of growing period, namely air temperature and precipitation. |
||
Comment 18. Conclusions: The conclusion should focus more on practical guidance for agricultural production. Response 18: Conclusions were supplemented with practical guidance for agricultural production (line s407-412). |
Reviewer 2 Report
Comments and Suggestions for Authors
Dear Authors, you should address my comments highlighted across the text.

Author Response
Response to Reviewer 2 Comments |
|||||||||||||||||
1. Summary |
|
|
|||||||||||||||
Thank you very much for taking the time to review this manuscript. Please find the detailed responses below and the corresponding revisions/corrections highlighted/in track changes in the re-submitted files. |
|||||||||||||||||
2. Point-by-point response to Comments and Suggestions for Authors |
|||||||||||||||||
Comment 1: Suggestions for changes in wording in Abstract. Response 1: The wording suggested by the reviewer has been changed accordingly in Abstract. |
|||||||||||||||||
Comment 2: Please, replace this word with a new one not included in the title. Response 2: Keyword “phytate” (line 33) has been changed into a new word (phytic acid) not included in the title. |
|||||||||||||||||
Comment 3: The experimental protocol should be described making it clearer the experimental factors applied with the related levels, as well as the design used for the treatment distribution in the field. Response 3: Tabel 1. Scheme of the field experiment organic (Org0-Org2) and conventional (N1-N3) treatments with replications (I-IV) was added to the Material and Methods section.
|
|||||||||||||||||
Comment 4: I would recommend to replace the word 'variant' across the text with a more appropriate term within the statistical frame. Response 4: We appreciate the suggestion, as term “variant” is not precise when talking about crops grown within the experiment. The term across the text was removed or replaced with the word “treatment”. |
|||||||||||||||||
Comment 5: . Response 5: Dot was added to the end of the title of Table 2. |
|||||||||||||||||
Comment 5: . Response 5: Dot was added to the end of the title of Table 2. |
|||||||||||||||||
Comment 6: The first letter of each title and sub-title must be capital. Response 6: Changes to the title and sub-titles was applied. |
|||||||||||||||||
Comment 7: . Response 7: Dot was added to the end of the title of Figure 1. |
|||||||||||||||||
Comment 8: Suggestions for changes in wording in Conclusions. Response 8: The wording suggested by the reviewer has been changed accordingly on Conclusion. |